# The mitochondrial acyl carrier protein (ACP) coordinates mitochondrial fatty acid synthesis with iron sulfur cluster biogenesis

Jonathan G Van Vranken[1†], Mi-Young Jeong[1,2†], Peng Wei[1], Yu-Chan Chen[1‡], Steven P Gygi[3], Dennis R Winge[1,2*], Jared Rutter[1,4*]

[1]Department of Biochemistry, University of Utah School of Medicine, Salt Lake City, United States; [2]Department of Medicine, University of Utah School of Medicine, Salt Lake City, United States; [3]Department of Cell Biology, Harvard University School of Medicine, Boston, United States; [4]Howard Hughes Medical Institute, University of Utah School of Medicine, Salt Lake City, United States

*For correspondence: dennis.
winge@hsc.utah.edu (DRW);
rutter@biochem.utah.edu (JR)

†These authors contributed
equally to this work

Present address: ‡Department
of Biology, Stanford University,
Stanford, United States

Competing interests: The
authors declare that no
competing interests exist.

Reviewing editor: J Wade
Harper, Harvard Medical School,
United States

**Abstract** Mitochondrial fatty acid synthesis (FASII) and iron sulfur cluster (FeS) biogenesis are both vital biosynthetic processes within mitochondria. In this study, we demonstrate that the mitochondrial acyl carrier protein (ACP), which has a well-known role in FASII, plays an unexpected and evolutionarily conserved role in FeS biogenesis. ACP is a stable and essential subunit of the eukaryotic FeS biogenesis complex. In the absence of ACP, the complex is destabilized resulting in a profound depletion of FeS throughout the cell. This role of ACP depends upon its covalently bound 4'-phosphopantetheine (4-PP)-conjugated acyl chain to support maximal cysteine desulfurase activity. Thus, it is likely that ACP is not simply an obligate subunit but also exploits the 4-PP-conjugated acyl chain to coordinate mitochondrial fatty acid and FeS biogenesis.

## Introduction

The mitochondrial acyl carrier protein (ACP; *Figure 1—figure supplement 1*) plays a critical role in the evolutionarily conserved type II fatty acid biosynthesis pathway (FASII; *Figure 1—figure supplement 2A*). Unlike the cytosolic fatty acid biosynthesis pathway (FASI), the mitochondrial FASII system, which is homologous to the prokaryotic fatty acid biosynthesis pathway, utilizes a set of monofunctional enzymes that interact transiently with ACP to catalyze the initiation and elongation of nascent acyl chains (*Hiltunen et al., 2010*). To facilitate FASII, ACP utilizes a 4'-phosphopantetheine prosthetic group (4-PP), which is covalently bound to an invariant Ser residue (*Majerus et al., 1965*; *Stuible et al., 1998*). As such, ACP serves as a soluble scaffold for acyl intermediates during the stepwise process of *de novo* fatty acid synthesis. Currently, it is thought that the primary product of ACP-dependent FASII is octanoate, which is cleaved from ACP and further processed to generate lipoic acid. Lipoic acid is an obligate cofactor of the pyruvate dehydrogenase and α-ketoglutarate dehydrogenase complexes as well as the branched chain α-keto acid dehydrogenase and glycine cleavage complex (*Hiltunen et al., 2010*; *Brody et al., 1997*). In addition, a FASII-derived acyl chain other than lipoic acid is required for RNase P function in tRNA maturation (*Schonauer et al., 2008*).

Biochemical analyses of mammalian FASII enzymes demonstrate that this pathway is capable of generating ACP-bound acyl chains as long as fourteen carbons (*Zhang et al., 2005*). Since lipoic acid biosynthesis requires an acyl chain of just eight carbons, it is likely that these extended FASII-synthesized fatty acids serve an alternative function in mitochondria (*Brody et al., 1997*). Indeed,

**eLife digest** Like animals and plants, yeast cells contain structures called mitochondria. These structures are commonly referred to as the powerhouses of the cell because they provide much of the energy that cells need to survive. All mitochondria contain a protein called acyl carrier protein (ACP), which cells need in order to live. The ACP protein has a number of known roles including manufacturing the molecules that make up certain fats and helping to organise other proteins that are important for energy production. However, neither of these roles explain why yeast cells require ACP because the other proteins required for these processes are not required for survival.

Mitochondria are also the sites where iron and sulfur atoms are joined together to make the iron sulfur clusters that many proteins need in order to carry out their roles. Van Vranken, Jeong et al. now show that the ACP protein associates with a molecular machine that makes iron sulfur clusters in the mitochondria of budding yeast cells. The experiments show that this interaction is needed to produce iron sulfur clusters, and without it the other proteins involved in the process are not able to work together. Since iron sulfur clusters are essential for life, this could explain why cells cannot survive without ACP. Van Vranken et al. also showed that ACP is only able to efficiently produce iron sulfur clusters when a chemical called a "4-PP-conjugated acyl chain" is attached to it.

It is possible to separate the activity of ACP in making iron sulfur clusters from its previously known roles. Van Vranken et al. suggest that the addition of the 4-PP-conjugated acyl chain to ACP may help to balance the use of ACP between its different activities. Moving forward, Van Vranken et al. hope to determine the structure of ACP in more detail to understand how it contributes to iron sulfur cluster formation, and why this single protein has evolved to perform so many distinct roles.

proteomic and structural studies have demonstrated that ACP is a stable accessory subunit of mitochondrial respiratory Complex I (CI) (*Sackmann et al., 1991*; *Angerer et al., 2014*). Furthermore, the pool of ACP associated with CI contains a 4-PP-conjugated 3-hydroxymyristic acid, however, the functional importance of this 14-carbon acyl chain has never been investigated in the context of CI activity or assembly (*Carroll et al., 2003*).

In *Saccharomyces cerevisiae*, *ACP1*, the gene encoding ACP, is essential for cell viability in multiple strain backgrounds, while the genes required for lipoic acid biosynthesis and ligation are not (*Figure 1A*, *Figure 1—figure supplement 2B,C*) (*Schonauer et al., 2008*; *Brody et al., 1997*). This is consistent with reports demonstrating that ACP is essential for viability in *Yarrowia lipolytica* (*Dobrynin et al., 2010*) and in mammalian cells (*Yi and Maeda, 2005*; *Feng et al., 2009*). Moreover, *S. cerevisiae* mitochondria do not have CI or any structurally similar analog of it. Since neither known function explains the essentiality of *ACP1*, we reasoned that ACP must perform a distinct, unknown, and essential mitochondrial function.

## Results and conclusions

To begin to define the essential function of Acp1, we purified endogenously expressed and fully functional Acp1-HA (*Figure 1A*) from purified mitochondria to discover interacting proteins that might explain the *acp1Δ* phenotype. We were particularly intrigued by the co-purification of three subunits of the ISU complex—Nfs1, Isd11, and Isu1—each of which is required for FeS biogenesis and essential for viability. The cysteine desulfurase (Nfs1) and Isd11 form the core of the ISU complex and catalyze the conversion of cysteine to alanine thereby generating a persulfide intermediate, which is the source of sulfide ions that combine with ferrous iron on the Isu1 scaffold protein to form [2Fe-2S] clusters (*Garland et al., 1999*; *Mühlenhoff et al., 2004*; *Adam et al., 2006*; *Wiedemann et al., 2006*). To confirm these interactions, we immunoprecipitated Acp1-HA from isolated mitochondria and analyzed the eluates by immunoblot. Indeed, Nfs1, Isd11, and Isu1 all specifically co-immunoprecipitate with Acp1, although Isu1 appears to interact less avidly than Nfs1 or Isd11 (*Figure 1B*). In addition to SDS-PAGE, the resultant eluates were also resolved by blue native (BN)-PAGE, which demonstrated that Acp1 co-purifies with the intact core Nfs1-Isd11 complex (*Figure 1B*). Finally, Nfs1-V5 and Isd11-V5 were each immunoprecipitated from isolated

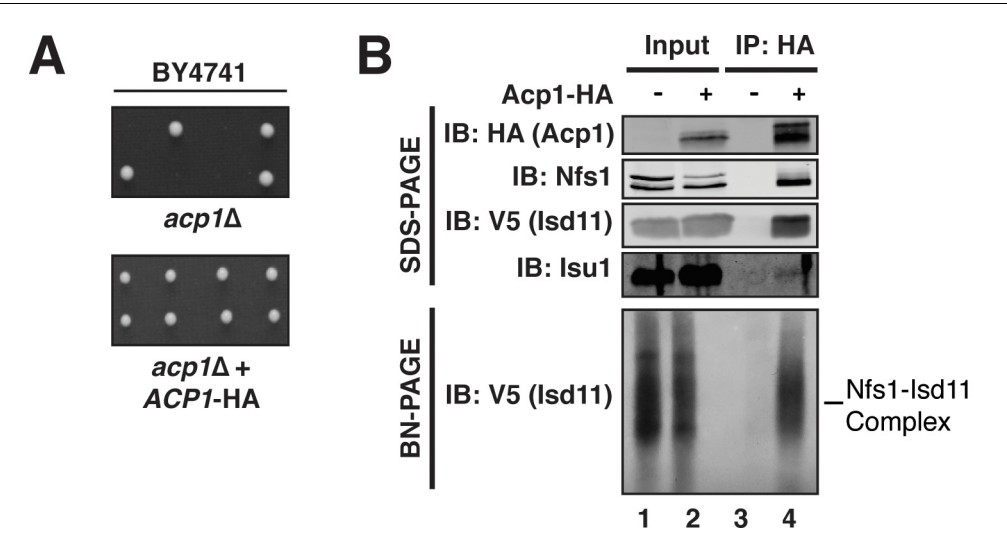

**Figure 1.** Acp1 is a stable subunit of the ISU complex. (**A**) *acp1Δ* heterozygous diploids were dissected with and without a plasmid expressing Acp1-HA (BY4741) and spores were grown on YPAD medium for 2 days. Sporulation of the heterozygous deletion strain failed to generate haploid *ACP1* deletion strains unless a vector borne *ACP1* gene was present. (**B**) Purified mitochondria from cells either expressing Acp1-HA or not were solubilized by digitonin (input) and then subjected to anti-HA immunoprecipitation. The resulting eluates and input samples were subjected to SDS-PAGE and BN-PAGE and immunoblot.

The following figure supplements are available for figure 1:

**Figure supplement 1.** Protein sequence alignment of ACP from eukaryotes.

**Figure supplement 2.** Eukarotic mitochondrial fatty acid biosynthesis (FASII) pathway.

**Figure supplement 3.** Acp1 is a stable subunit of the ISU complex.

mitochondria. As expected, Acp1-HA was detected in the eluates of each immunoprecipitation (*Figure 1—figure supplement 3*). These results are further supported by proteomics-based interaction studies, which identify human ACP, NFS1, and ISD11 as mutually interacting proteins in mammalian cells (*Huttlin et al., 2015*). Taken together these data demonstrate that Acp1 is a stable and evolutionarily conserved subunit of the ISU complex with Nfs1, Isd11, and Isu1.

FeS biogenesis is an essential function of mitochondria and is absolutely dependent on the ISU complex with which Acp1 stably interacts (*Lill et al., 1999*). Thus, we hypothesized that the essential function of Acp1 might relate to FeS biogenesis. We engineered inducible *ACP1* knockdown strains (Acp1[KD]) using two distinct strategies and strain backgrounds—TetO7-*ACP1* in the BY4741 background and Gal-*ACP1* in the DY150 background, in which *ACP1* expression is suppressed by doxycycline and galactose withdrawal, respectively. As expected, Acp1[KD] cells from each background displayed attenuated growth upon *ACP1* shutdown, which is particularly evident on respiration-requiring glycerol medium (*Figure 2—figure supplement 1A–C*). Importantly, viability could be restored in each of these strains by episomal expression of the Acp1-HA at endogenous levels. In addition to the expected loss of lipoic acid-containing subunits of pyruvate dehydrogenase and α-ketoglutarate dehydrogenase in the Acp1[KD] cells, we also observed a specific destabilization of the FeS-containing subunits of Complex II (Sdh2) and Complex III (Rip1) and loss of those assembled respiratory complexes (*Figure 2A* and *Figure 2—figure supplement 2A*). A similar destabilization of these complexes occurs in cells depleted for Nfs1 and Isu1 (*Adam et al., 2006*; *Wiedemann et al., 2006*). Importantly, Complex III biogenesis stalls in Acp1[KD] cells at the final stage of assembly – incorporation of the Rieske FeS protein (Rip1) (*Figure 2A*; IB: Rip1)–resulting in the accumulation of a stable assembly intermediate of Complex III lacking Rip1 (*Figure 2A*; III$_2$*; IB:

Qcr7). The presence of the late stage intermediate $III_2^*$ indicates that mitochondrial translation of the Cob cytochrome b subunit is normal in $Acp1^{KD}$ cells (*Atkinson et al., 2011*; *Cui et al., 2012*). Likewise, translation of the mitochondrial subunits of ATP synthase is normal as seen by the assembled $F_1F_0$ complex (*Figure 2A*). We also observed a loss of activity of aconitase, a mitochondrial enzyme with an obligate FeS cofactor (*Figure 2B* and *Figure 2—figure supplement 2B*).

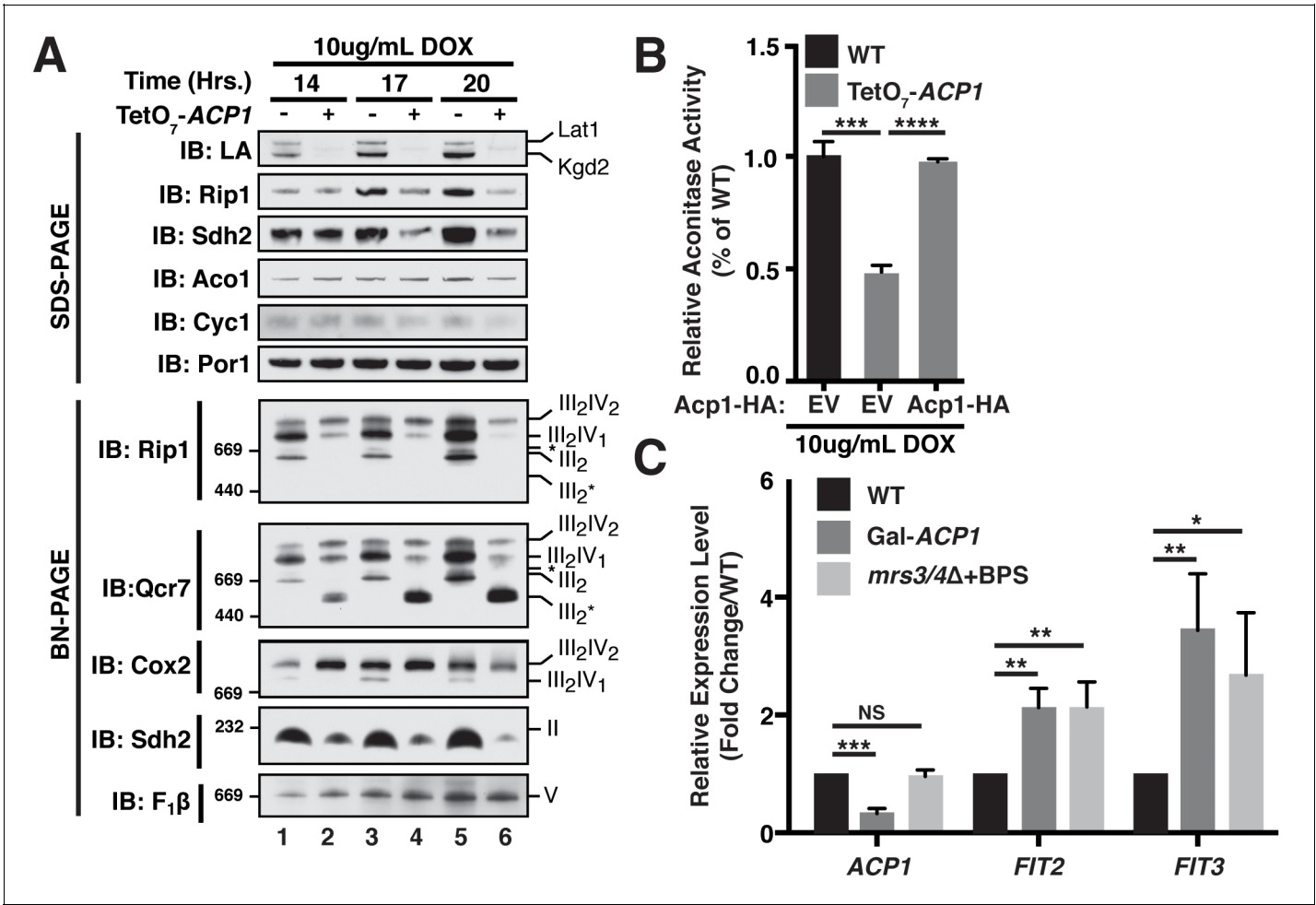

**Figure 2.** Acp1 is required for FeS biogenesis. (**A**) Isolated mitochondria from the indicated strains were resolved by SDS-PAGE (upper panel) or solubilized in 1% digitonin and resolved by BN-PAGE (lower panel). The time course indicates the time following addition of 10 µg/mL doxycycline to the cultures, which suppresses expression from the $TetO_7$-*ACP1* allele. The indicated proteins and protein complexes were assessed by immunoblot. LA indicates lipoic acid-conjugated Lat1 (PDH complex subunit; upper band) and Kgd2 (α-ketoglutarate dehydrogenase complex subunit; lower band). (**B**) Aconitase activity was measured in whole cell lysates from the indicated strains containing the indicated plasmids 18 hr post-addition of 10 µg/mL doxycycline (± SEM; N = 3 biological replicates. ***p<0.0005, ****p<0.00005). (**C**) qPCR was used to measure the expression of *ACP1*, *FIT2*, and *FIT3* in the indicated strains. The Gal-*ACP1* strain was harvested at 28 hr post-transfer to raffinose medium to suppress *ACP1* expression. The *mrs3/4Δ* strain lacks both Mrs3 and Mrs4 mitochondrial iron transporters. BPS (80 µM) is a Fe(II) chelator that causes depletion of bioavailable iron. *FIT2* and *FIT3* are components of the iron regulon that is induced upon loss of cytosolic FeS (± SEM; N = 3 biological replicates. *p<0.05, **p<0.005, ***p<0.0005).

The following source data and figure supplements are available for figure 2:

**Source data 1.** Source data for *Figure 2*.

**Figure supplement 1.** *ACP1* expression is required for cell proliferation.

**Figure supplement 2.** Acp1 is essential for FeS biogenesis.

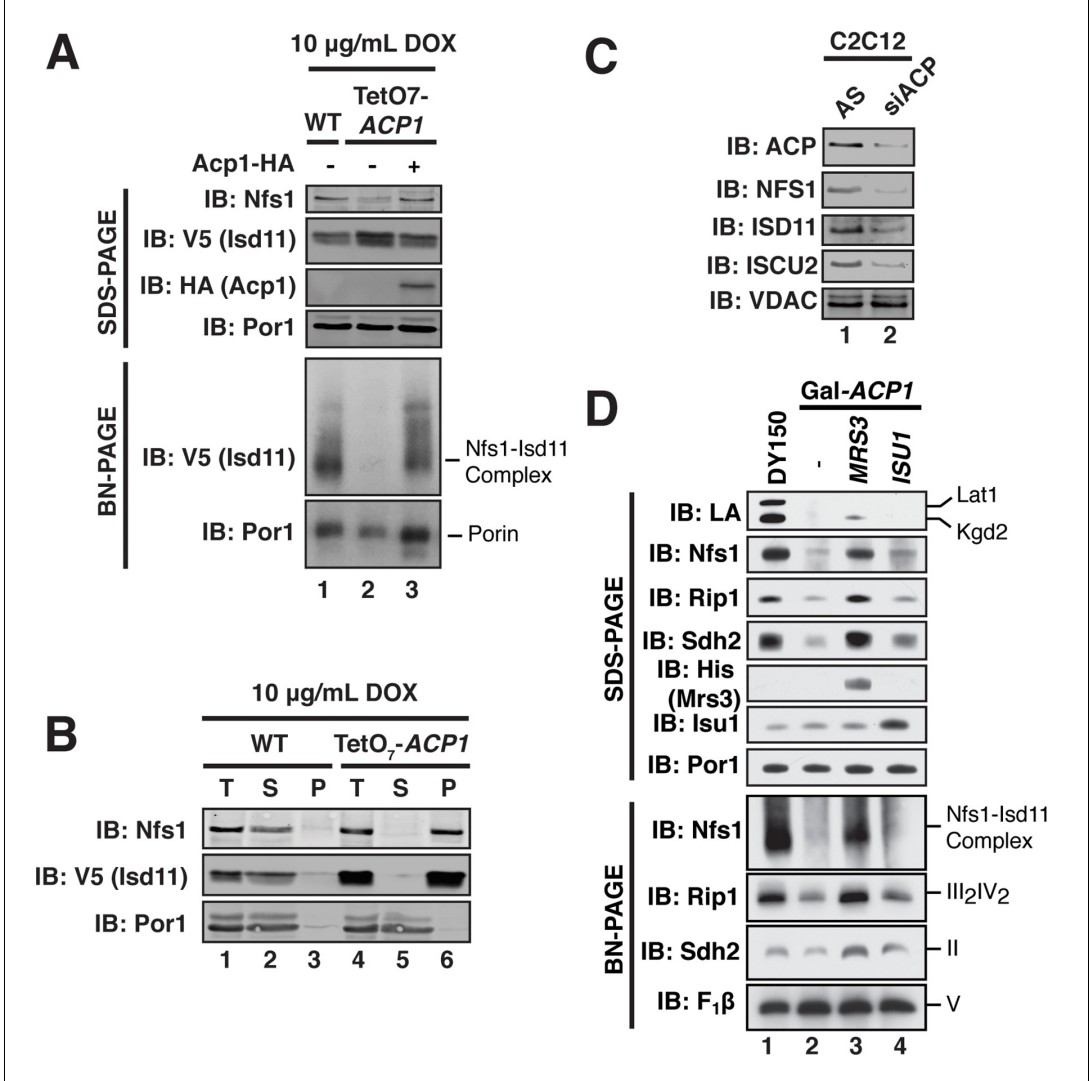

**Figure 3.** ACP promotes FeS biogenesis by maintaining the stability of the ISU (Nfs1-Isd11) complex. (**A**) Purified mitochondria from the indicated strains were either resolved by SDS-PAGE (lower panels) or solubilized in 1% digitonin and resolved by BN-PAGE (upper panels). Cells were grown for 18 hr in the presence of 10 µg/mL doxycycline. The indicated proteins and protein complexes were assessed by immunoblot. (**B**) Mitochondria purified from the indicated strains 18 hr post-addition of 10 µg/mL doxycycline were solubilized with 1% Triton X-100. Soluble (S) and pellet (P) fractions were separated by centrifugation at 100,000 $g$. The fractions, along with the total input (T) were resolved by SDS-PAGE and assessed by immunoblot. (**C**) C2C12 mouse myoblasts were transfected with a pool of siRNA targeting *NDUFAB1* (ACP) or a scrambled control. Isolated mitochondria was resolved by SDS-PAGE and assessed by immunoblot. (**D**) Isolated mitochondria from the WT and Gal-*ACP1* strains expressing the indicated gene via a 2 µ plasmid at 28 hr post-transfer to raffinose-containing medium were either resolved by SDS-PAGE (lower panels) or solubilized in 1% digitonin and resolved by BN-PAGE (upper panels). The indicated proteins and protein complexes were assessed by immunoblot.

The following source data and figure supplement are available for figure 3:

**Source data 1.** Source data for *Figure 3*.
**Figure supplement 1.** ACP promotes FeS biogenesis by maintaining the stability of the ISU (Nfs1-Isd11) complex.

The mitochondrial ISU complex is essential for the production of FeS that act in the cytosol as well as ribosome assembly (*Kispal et al., 2005*). In addition, mitochondrial FeS synthesis is important to attenuate the transcriptional activity of two partially redundant iron-responsive factors Aft1 and Aft2. In Acp1[KD] cells the expression of Aft1-target genes *FIT2* and *FIT3* was elevated consistent with impaired mitochondrial FeS synthesis (*Figure 2C*) (*Chen et al., 2004*; *Rutherford et al., 2005*).

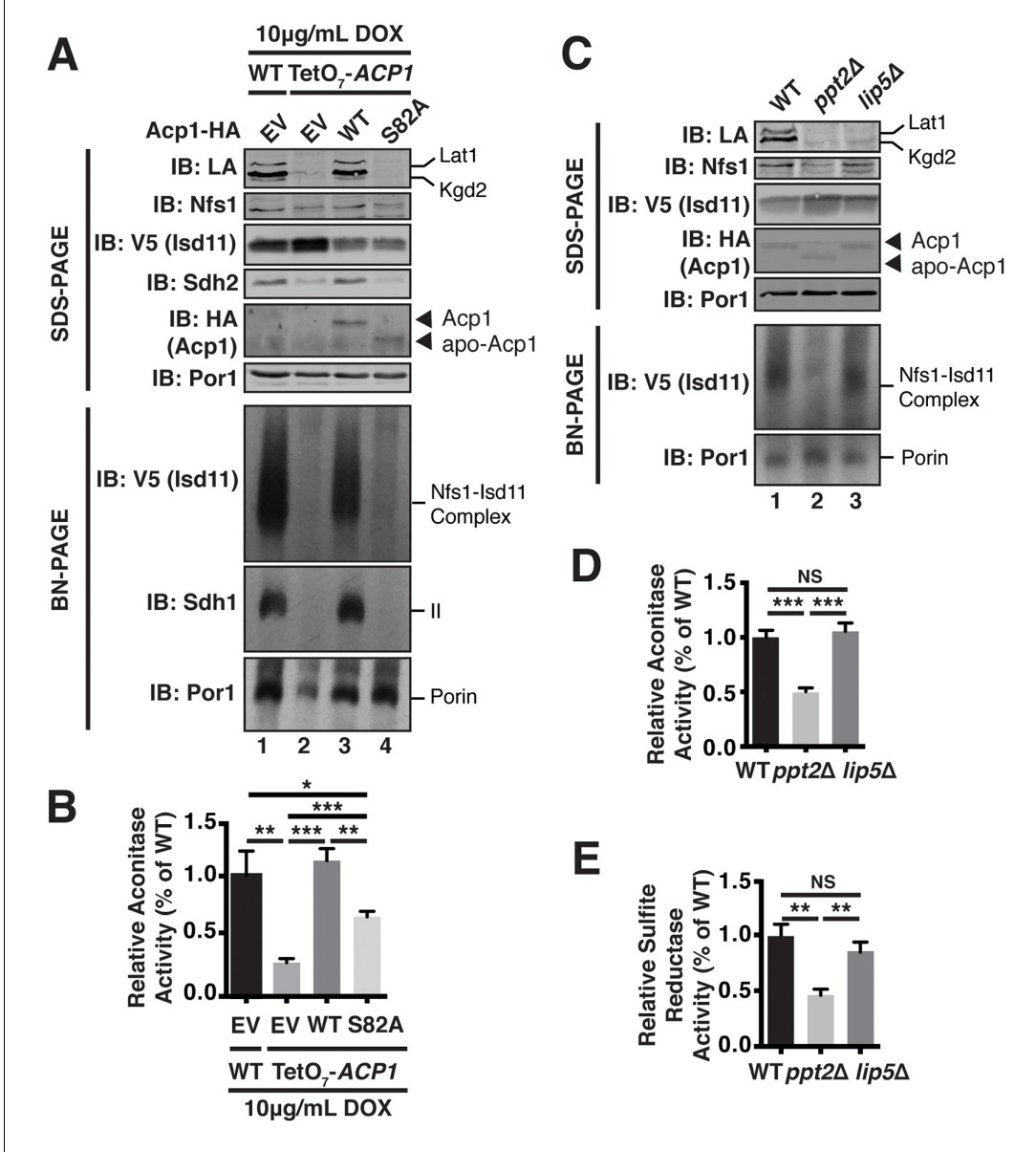

**Figure 4.** Acp1 requires a 4-PP-conjugated acyl chain to fully stabilize the ISU complex. (**A**) Isolated mitochondria from the indicated strains expressing the indicated genes by plasmid were either resolved by SDS-PAGE (upper panels) or solubilized in 1% digitonin and resolved by BN-PAGE (lower panels). The indicated proteins and protein complexes were assessed by immunoblot. Cells were grown for 18 hr in the presence of 10 μg/mL doxycycline. (**B**) Aconitase activity was measured in whole cell lysates from the indicated strains grown for 18 hr in the presence of 10 μg/mL doxycycline (± SEM; N = 3 biological replicates. *p<0.05, **p<0.005, ***p<0.0005). (**C**) Isolated mitochondria from the indicated strains were either resolved by SDS-PAGE (upper panels) or solubilized in 1% digitonin and resolved by BN-PAGE (lower panels). The indicated proteins and protein complexes were assessed by immunoblot. (**D**) Aconitase activity was measured in whole cell lysates from the indicated strains (± SEM; N = 3 biological replicates. ***p<0.0005). (**E**) Sulfite reductase activity was measured in whole cell lysates from the indicated strains (± SEM; N = 3 biological replicates. **p<0.005).

The following source data and figure supplement are available for figure 4:

**Source data 1.** Source data for *Figure 4*.

**Figure supplement 1.** Acp1 requires a 4-PP-conjugated acyl chain to fully stabilize the ISU complex.

These Aft1 target genes are also induced in cells lacking the two mitochondrial iron transporters Mrs3 and Mrs4 (*Figure 2C*). To further assess the perturbation of cytosolic FeS function, we quantified the activity of the cytosolic FeS-containing enzyme sulfite reductase and observed a diminution in Acp1$^{KD}$ cells (*Figure 2—figure supplement 2C*). Combined with the essential nature of *ACP1*, these data demonstrate that Acp1 is essential for FeS biogenesis.

We next sought to define the mechanism underlying the observed necessity of Acp1 for FeS biogenesis, focusing specifically on the ISU complex. Acp1$^{KD}$ cells exhibited a marked diminution of the assembled Nfs1-Isd11 complex, similar to depletion of the other ISU complex subunits (*Figure 3A*, *Figure 3—figure supplement 1A,B*). Importantly, the Nfs1 and Isd11 protein that remain in these cells is found only in insoluble aggregates in contrast to WT cells where Nfs1 and Isd11 are soluble (*Figure 3B*). Thus, in the absence of Acp1, the Nfs1-Isd11 complex is destabilized, most likely causing a loss of cysteine desulfurase activity. To determine if mammalian ACP is also necessary for maintaining steady state levels of the FeS biogenesis complex in mammalian cells, C2C12 mouse myoblasts were transfected with a pool of siRNAs targeting *NDUFAB1* (the gene encoding mammalian ACP) or a control siRNA. While our methods were unable to detect the mammalian Nfs1-Isd11 complex by BN-PAGE, depletion of ACP in these cells was accompanied by a clear destabilization of the subunits of the mammalian FeS biogenesis machinery, NFS1, ISD11, and ISCU2, which is the mammalian version of Isu1 (*Figure 3C*). Thus, ACP plays an evolutionarily conserved role in stabilizing the ISU complex thereby enabling FeS biogenesis.

Interestingly, overexpression of the *MRS3* iron transporter resulted in robust stabilization of the Nfs1-Isd11 complex in Acp1$^{KD}$ cells, while not restoring lipoic acid biosynthesis (*Figure 3D*). Elevated Mrs3 expression also restored Sdh2 and Rip1 protein abundance and complex assembly as well as aconitase activity in Acp1$^{KD}$ cells (*Figure 3D* and *Figure 3—figure supplement 1C*). We tested whether overexpression of other components of the ISU complex stabilized the Nfs1-Isd11 complex in cells depleted of Acp1. Elevated levels of Isu1 yielded a modest stabilization and restored aconitase activity (*Figure 3D* and *Figure 3—figure supplement 1C*), while overexpression of Yfh1, the yeast frataxin homologue, had no effect (*Figure 3—figure supplement 1D*). We speculate that elevated Mrs3 may increase the Fe(II) occupancy of Isu1, which enables it to more effectively stabilize the ISU complex.

Acp1 requires a 4-PP prosthetic group to support mitochondrial fatty acid synthesis. The gene encoding 4-PP transferase, *PPT2*, is not essential, but the haploid deletion strain exhibits no growth in respiration-requiring medium and impaired growth on glucose, which does not require respiration (*Figure 1—figure supplement 2C* and *Figure 4—figure supplement 1A*). The growth impairment on glucose is not explained by any known function of ACP and therefore may relate to defects in FeS biogenesis.

To directly test the role of the 4-PP prosthetic group and the acyl chain that is conjugated to it, we investigated the ability of apo-Acp1 lacking 4-PP to support FeS biogenesis. Apo-Acp1 can be generated in vivo by mutating the invariant Ser (S82) to which 4-PP is conjugated (*Stuible et al., 1998*). While re-expression of WT Acp1 could fully restore the steady state abundance of the Nfs1-Isd11 complex in Acp1$^{KD}$ cells, expression of Acp1$^{S82A}$ had only modest effects (*Figure 4A*). These modest effects were sufficient to enable the Acp1$^{KD}$ cells expressing Acp1$^{S82A}$ to retain viability, albeit with impaired grow rate, and to exhibit a modest recovery of aconitase activity compared to Acp1$^{KD}$ cells (*Figure 4B*, *Figure 4—figure supplement 1B*). Thus, the apparent absence of the Nfs1-Isd11 complex on BN gels is likely the result of a severely destabilized complex that is not capable of surviving the stringent detergent conditions of BN-PAGE and not the complete loss of the complex in vivo.

To further investigate the role of the 4-PP-conjugated acyl chain in FeS biogenesis we interrogated the effects of *PPT2* deletion on the function of Acp1 in FeS biogenesis. Like Acp1$^{KD}$ cells expressing Acp1$^{S82A}$, the Nfs1-Isd11 complex was severely depleted in *ppt2Δ* cells (*Figure 4C*). Furthermore, these cells exhibit a clear diminution in activity of FeS-containing enzymes in both the mitochondria and cytosol as represented by aconitase and sulfite reductase activity, respectively (*Figure 4D,E*). We also interrogated the ability of Acp1 to interact with Nfs1 in WT and *ppt2Δ* cells. While the steady state levels of Acp1 are not affected in *ppt2Δ* cells (*Figure 4C*), we observed a clear defect in the ability of Acp1 to interact with Nfs1 in this strain (*Figure 4—figure supplement 1C*). Therefore, the Acp1-conjugated 4-PP plays an important role in the interaction of Acp1 with the core Nfs1-Isd11 complex and in FeS biogenesis. Importantly, *lip5Δ* cells, which cannot synthesize

lipoic acid but remain competent for Acp1-dependent fatty acid synthesis (*Hiltunen et al., 2010*), maintain normal abundance of the Nfs1-Isd11 complex and aconitase and sulfide reductase activity (*Figure 4C–E*). Thus, the defects in FeS biogenesis observed in cells expressing apo-Acp1 are a result of the inability to generate an acyl-conjugated Acp1 species and not a defect in lipoic acid biosynthesis.

The data presented herein define a new and unexpected role of ACP in FeS biogenesis. ACP functions as a stable subunit of the ISU complex where it acts to stabilize the complex in part by exploiting a 4-PP-conjugated acyl chain. Unlike ACP, however, the acyl chain is not absolutely required for FeS biogenesis and viability, which raises the intriguing possibility that ACP is not simply an obligate subunit, but may exploit this unique interaction modality to provide additional structural or regulatory functions on FeS biogenesis. It is particularly intriguing to speculate that ACP may serve to coordinate mitochondrial fatty acid synthesis and FeS biogenesis, which represent two critical biosynthetic processes performed by mitochondria.

## Materials and methods

### Yeast strains and growth conditions

*Saccharomyces cerevisiae* BY4741 (*MATa, his3 leu2 met15 ura3*), *Saccharomyces cerevisiae* R1158 (BY4741 derivative; *MATa*, URA3::CMV-tTA, *his3 leu2 met15*), *Saccharomyces cerevisiae* W303a (*MATa, his3 leu2 met15 trp1 ura3*), and *Saccharomyces cerevisiae* DY150 (*MATa ade2-1 his3-11 leu2-3,112 trp1-1 ura3-52 can1-100(oc)*) were used as the wild-type strains where indicated. Each mutant was generated using a standard PCR-based homologous recombination method. The genotypes of all strains used in this study are listed in *Supplementary file 1*. Yeast transformation was performed by the standard TE/LiAc method and transformed cells were recovered and grown in synthetic complete glucose (SD) medium lacking the appropriate amino acid(s) for selection purposes. Medium used in this study includes YPA and synthetic minimal medium supplemented with 2% glucose, 2% raffinose, or 2% glycerol.

Growth assays were performed using synthetic minimal media supplemented with the appropriate amino acids and indicated carbon source. For plate-based growth assays, overnight cultures were back-diluted to equivalent ODs and spotted as 10-fold serial dilutions. For liquid culture growth assays, overnight cultures were back-diluted to equivalent ODs and grown at 30°C. Growth was monitored by absorbance at 600 nm.

To shut down expression of *ACP1* in TetO$_7$-*ACP1*, over-night cultures were used to inoculate synthetic media containing either 2% glucose or 2% raffinose and 10 µg/mL DOX to an approximate OD$_{600}$ of 0.05 and incubated for 16–24 hr as indicated. To shut down the expression in Gal-*ACP1*, Gal-*NFS1*, Gal-*ISD11*, Gal-*ISU1*, and Met3-*YFH1*, over-night cultured cells were used to inoculate in synthetic media containing 2% raffinose to an approximate OD$_{600}$ of 0.05 and incubated from 24 to 32 hr as indicated. For *YFH1* shut down 2.5 mM methionine was added in the media.

### Isolation of yeast mitochondria

Cell pellet was washed once with ddH$_2$O and incubated in TD buffer (100 mM Tris-SO$_4$, pH 9.4 and 100 mM DTT) for 15 min at 30°C. Spheroplasts were obtained by incubating cells in SP buffer (1.2 M Sorbitol and 20 mM potassium phosphate, pH 7.4) supplemented with 0.3 mg/mL lyticase for 1 hr at 30°C to remove the cell wall. Spheroplasts were gently washed once and homogenized in ice-cold SEH buffer (0.6 M sorbitol, 20 mM HEPES-KOH, pH 7.4, 2 mM MgCl$_2$, 1 mM EGTA) using a dounce homogenizer applied with 30–40 strokes. Crude mitochondria were then isolated by differential centrifugation.

### Immunoprecipitation

Crude mitochondria were isolated and resuspended to a concentration of 5 mg/mL. Mitochondria was solubilized in 0.7% digitonin for 30 min. Followed by centrifugation at 20,000 ×g for 20 min. Cleared mitochondrial lysates were incubated with anti-HA antibody conjugated agarose (Sigma) for 2 hr. at 4°C. The agarose was washed 3–5 times and eluted in Laemmli buffer (65°C, 10 min). Elutions were resolved by SDS-PAGE and assessed by immunoblot.

## Steady-state protein analysis

Yeast mitochondria were solubilized in Laemmli buffer. Samples were resolved by SDS-PAGE and assessed by immunoblot.

## Blue native polyacrylamide gel electrophoresis (BN-PAGE)

BN-PAGE was performed as described previously (*Wittig et al., 2006*). Mitochondria were resuspended in lysis buffer (Invitrogen) and solubilized with 1% digitonin. Lysates were resolved on a 4%–16% gradient native gel (Invitrogen).

## Protein aggregation assays

Mitochondria were solubilized in Triton-X100 lysis buffer (0.5% Triton-X100, 20 mM HEPES-KOH, pH 7.4, 150 mM KCl). The samples were incubated on ice for 30 min. and centrifuged at 30,000 ×g for 10 min.

## Aconitase activity assays

Yeast cells were grown in SD medium to early log phase, resuspended in lysis buffer (50 mM Tris-HCl, 50 mM KCl, 2 mM sodium citrate dihydrate, 10% glycerol, 1 mM PMSF, and 7 mM β-mercaptoethanol), and stored at –80°C overnight. After thawing on ice, cells were homogenized by vortexing with glass beads and cleared lysate was collected by centrifugation. Aconitase activity was measured by coupling with $NADP^+$- dependent isocitrate dehydrogenase activity. 30 µl of crude lysate was mixed with 150 µl of reaction mixture (1 M Tris-Cl pH 8.0, 10 mM $MgCl_2$, 10 mM $NADP^+$, 0.32 units of $NADP^+$- dependent Isocitrate Dehydrogenase), and 10 µl of 50 mM citrate. The reaction mixture was recorded at 340 nm for 2 min (15 s intervals). Aconitase activity was normalized to total protein concentration.

## Sulfite reductase assays

To measure cytosolic iron, cells (50 ml cultures) were grown in SC –Met (To avoid repression of the enzyme expression by methionine) medium containing 2% glucose or raffinose as a carbon source till 1 of $OD_{600\ nm}$. Total cell lysate preparation and the enzyme assay was performed as described in (*Rutherford et al., 2005*) with modifications. Briefly, cell pellets were resuspended in buffer A (0.1 M Tris-Cl pH7.4, 10% glycerol, 1 mM EDTA pH 8.0, 1 mM phenylmethlysulfony fluoride (PMSF)) with lyticase and incubated 30°C for 45 min. After disruption using glass beads, 50 µl of cell lysates were mixed with 400 µl of assay mix with or without sulfite. After incubation at 37°C for 20 min, 100 µl of N,N-diethyl-p-phenylenediamine sulfate (DPD) and 100 µl of ferric chloride were added to the reaction mix to stop the reaction and incubated in the dark to develop the color for 20 min. The production of methylene blue was measured at 669 nm.

## qPCR analysis

To quantify the expression of Fe regulon genes, total RNAs were extracted from yeast spheroplasts using RNeasy mini kit (QIAGEN). cDNA were synthesized from 1 µg of total RNA using High-Capacity cDNA Reverse Transcription Kit (Applied Biosystems). 2 µl of 10X diluted cDNA reaction mix were mixed with SYBR Green real-time PCR master mix (Thermo Fisher) with primers and the PCR reaction were performed using the Mastercycler ep *realplex* (Eppendorf). Expression of genes of interesting was normalized to actin and fold changes was analyzed using the $2^{-\Delta\Delta Ct}$ method.

Primers used (Primers were designed to have 60°C of Tm using Primer3Plus online program)
ACT1_For ATTATATGTTTAGAGGTTGCTGCTTTGG
ACT1_Rev CAATTCGTTGTAGAAGGTATGATGCC
FIT2_For ACAAAGGTTGTCACCGAAGG
FIT2_Rev GATGATTCGACGGCTTGAGT
FIT3_For TCCGCTTTGGTTCTATCTGC
FIT3_Rev AGTGCTGCTGGCGTAAGAGT
ACP1_For ACTCTCCCAACATTGCCAAC
ACP1_Rev CAGCCACTTTGTCAGGGATT

## Statistics

PRISM software was used to graph all quantitative data and perform statistical analyses. p values for pairwise comparisons were determined using a Student's t test.

## Acknowledgements

This work was supported by RO1GM110755 (to DRW and JR). JR is an Investigator of the Howard Hughes Medical Institute. Plasmids and galactose-regulated gene strains were generously provided by Dr. Andrew Dancis, Dr. Roland Lill and Dr. Jerry Kaplan.

## Additional information

### Funding

| Funder | Grant reference number | Author |
| --- | --- | --- |
| Howard Hughes Medical Institute | | Jared Rutter |
| National Institute of General Medical Sciences | RO1GM110755 | Dennis R Winge Jared Rutter |

The funders had no role in study design, data collection and interpretation, or the decision to submit the work for publication.

### Author contributions

JGV, Conception and design, Acquisition of data, Analysis and interpretation of data, Drafting or revising the article; M-YJ, Acquisition of data, Analysis and interpretation of data, Drafting or revising the article; PW, Acquisition of data, Analysis and interpretation of data; Y-CC, Acquisition of data; SPG, Analysis and interpretation of data, Contributed unpublished essential data or reagents; DRW, JR, Conception and design, Analysis and interpretation of data, Drafting or revising the article

### Author ORCIDs

Jonathan G Van Vranken, http://orcid.org/0000-0002-8931-852X
Dennis R Winge, http://orcid.org/0000-0003-1160-1189
Jared Rutter, http://orcid.org/0000-0002-2710-9765

## Additional files

### Supplementary files

• Supplementary file 1. Yeast strains used in this study. This table describes the name, genotype, and source of all yeast strains used in this investigation.

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
