## [Decision Letter]

Thank you for submitting your article "The mitochondrial acyl carrier protein (ACP) coordinates mitochondrial fatty acid synthesis with FeS cluster biogenesis" for consideration by *eLife*. Your article has been reviewed by two peer reviewers, and the evaluation has been overseen by a Reviewing Editor and Michael Marletta as the Senior Editor. The following individuals involved in review of your submission have agreed to reveal their identity: John Markley (Reviewer #2).

The reviewers have discussed the reviews with one another and the Reviewing Editor has drafted this decision to help you prepare a revised submission.

Summary:

In this paper, the authors find that in addition to its role as a scaffold for acyl chain elongation, Acp1 (*S. cerevisiae* homolog) is also a subunit of the ISU complex involved in FeS biogenesis, a result that was also recently reported in human cells in the context of a large scale interactome study. Using genetic tools in yeast, the authors demonstrate loss of the ISU complex and FeS biogenesis (specifically OXPHOS complexes II and III subunits), as well as loss of a lipoic acid-containing PDH subunit as expected. Cytosolic FeS-containing systems were also affected and convincingly demonstrated (qPCR of transcripts known to be upregulated upon impaired mito FeS synthesis, activity of cytosolic sulfite reductase). Turnover of mammalian ISU subunits upon knockdown of NDUFAB1 (human ACP) was also demonstrated, suggesting the phenomenon is conserved (however more work will be needed to establish this given the complexities of the dually and now potentially triple localized human ACP, although this is probably out of the scope of this short report). Finally, using a mutant of ACP unable to conjugate the acyl chain, the authors demonstrate its importance in lipoic acid synthesis as expected, as well as FeS biogenesis. Interestingly, cells expressing only ACP unable to conjugate the acyl chain remain viable, suggesting the essentiality of ACP is in fact due to its involvement in FeS biogenesis. Knockout of the enzyme conjugating the acyl chain to ACP had a similar effect. The authors also found that overexpression of a mitochondrial ion transporter and to a lesser extent another subunit of the ISU complex, restored the levels of the ISU complex (and substrates). Taken together the assumption is that the stability of the ISU complex is being impaired or increased respectively, and also that the acyl chain has a primarily structural role in the ISU complex rather than a functional one. This is very nicely controlled report demonstrating a totally unexpected role for ACP in FeS biogenesis.

Essential revisions:

The reviewers felt that the paper is very strong but there was a consensus that a stronger caveat should be made with respect to the possible role of the acyl carrier group, based on the phenotype of the S82 mutant. It is possible that there is exclusively a structural role but that there is a structural defect in the S82 mutant that reduces folding/assembly to some extent. This would require a textual change.

---

## [Author Response]

*Essential revisions:*

*The reviewers felt that the paper is very strong, and primarily minor changes are needed (see below).*

*There was a consensus that a stronger caveat should be made with respect to the possible role of the acyl carrier group, based on the phenotype of the S82 mutant. It is possible that there is exclusively a structural role but that there is a structural defect in the S82 mutant that reduces folding/assembly to some extent. This would require a textual change.*

We agree with the critique and have no direct evidence to prove that apo-Acp1 folds analogously to holo-Acp1 however we have several pieces of data that suggest the defects observed in the S82A mutant can be attributed to the absence of a covalently bound acylated 4’-phosphopantetheine cofactor. While this alone does not prove that the S82A mutant achieves a native confirmation, we also know that apo-Acp1 has wild-type stability as it accumulates to the same degree as holo-Acp1 and is fully soluble. Furthermore, it is not completely unable to bind Isd11 and Nfs1. Newly included data demonstrates that apo-Acp1 can still bind the core ISU complex albeit with much reduced affinity. This is consistent with the observations that ACP and ISD11 make protein-protein contacts independent of those mediated by the acylated –PP. Taken together we feel strongly that the defects observed in cells expressing Acp1^S82A^, which mirror the defects seen in *ppt2Δ* cells, stem from a failure to make essential cofactor-protein and lipid-protein contacts. So while this does not prove the lack of some misfolding, we feel that the data strongly supports a structural role for the acylated cofactor in the ISU complex. Therefore, we are reluctant to add a statement in the text that we are quite confident is wrong and will be strongly disproven in the next several months with the publication of the structure. However, if the reviewers still feel strongly about this point, we will, of course, include the caveat as a possible interpretation.

New data: Figure 4—figure supplement 1. In order to further interrogate the defects observed in cells expressing apo-Acp1, Acp1 was immunoprecipitated from WT and *ppt2Δ* cells. While Acp1 accumulated to similar extents and was readily soluble in each both strains, Acp1 in *ppt2Δ* cells pulled down far less Nfs1 than Acp1 from WT cells. This demonstrates that the 4-PP promotes, but is not absolutely essential, for mediating the Acp1-Nfs1 interaction.